# Alleviating the burden of diabetes with Health Equity Funds: Economic evaluation of the health and financial risk protection benefits in Cambodia

Isabelle Feldhaus[1]*, Somil Nagpal[2], Stéphane Verguet[1]

**1** Harvard T.H. Chan School of Public Health, Boston, MA, United States of America, **2** World Bank Group, Washington, DC, United States of America

* ifeldhaus@g.harvard.edu

**Data Availability Statement:** All data used in the model are reported in the article alongside their sources that are available in the published

## Abstract

In Cambodia, diabetes caused nearly 3% of the country's mortality in 2016 and became the fourth highest cause of disability in 2017. Providing sufficient financial risk protection from health care expenditures may be part of the solution towards effectively tackling the diabetes burden and motivating individuals to appropriately seek care to effectively manage their condition. In this study, we aim to estimate the distributional health and financial impacts of strategies providing financial coverage for diabetes services through the Health Equity Funds (HEF) in Cambodia. The trajectory of diabetes was represented using a Markov model to estimate the societal costs, health impacts, and individual out-of-pocket expenditures associated with six strategies of HEF coverage over a time horizon of 45 years. Input parameters for the model were compiled from published literature and publicly available household survey data. Strategies covered different combinations of types of diabetes care costs (i.e., diagnostic services, medications, and management of diabetes-related complications). Health impacts were computed as the number of disability-adjusted life-years (DALYs) averted and financial risk protection was analyzed in terms of cases of catastrophic health expenditure (CHE) averted. Model simulations demonstrated that coverage for medications would be cost-effective, accruing health benefits ($27 per DALY averted) and increases in financial risk protection ($2 per case of CHE averted) for the poorest in Cambodia. Women experienced particular gains in health and financial risk protection. Increasing the number of individuals eligible for financial coverage also improved the value of such investments. For HEF coverage, the government would pay between an estimated $28 and $58 per diabetic patient depending on the extent of coverage and services covered. Efforts to increase the availability of services and capacity of primary care facilities to support diabetes care could have far-reaching impacts on the burden of diabetes and contribute to long-term health system strengthening.

literature. See Methods and References for source details.

**Funding:** The author SN is an employee of the World Bank Group. The World Bank Group provided support in the form of salary for author SN, but the World Bank Group did not have any additional role in the study design, data collection and analysis, decision to publish, or preparation of the manuscript. The specific role of SN on this study is articulated in the 'author contributions' section. SV acknowledges funding support from the Trond Mohn Foundation and NORAD through BCEPS (#813596).

**Competing interests:** The author SN of this manuscript is an employee of the World Bank Group. This affiliation does not alter authors' adherence to all PLOS ONE policies on sharing data and materials.

## Introduction

### Health and economic burden of diabetes

Globally, the prevalence of diabetes among adults rose from 5% in 1980 to 9% in 2014 and has been rapidly rising in low- and middle-income countries (LMICs) [1,2]. In Cambodia, diabetes caused nearly 3% of the country's mortality in 2016 and became the fourth highest cause of disability in 2017 [3–6]. Prevalence of diabetes among women (3.3%) has also been reported to be higher than men (2.5%), and projections have estimated an 82% increase in the country's diabetes burden from 2008 to 2028 [6,7].

Despite high diabetes prevalence, most diabetics in the country are not aware of their condition. Most diabetics remain undiagnosed until the onset of severe complications and medications and treatments to manage the disease are frequently only available from private providers for those who can afford it [7,8]. Management of diabetes and its related complications is generally associated with high costs [9,10]. Furthermore, diabetes can disproportionately affect the poor with the potential to worsen financial stability and undermine poverty reduction efforts [4,11,12]. As health status, poverty, and economic development are closely linked, cost-effective strategies that provide sufficient financial access to essential health care will increasingly be needed.

### Diabetes care and management in Cambodia

Despite the adoption of the National Strategic Plan for the Prevention and Control of Non-communicable Diseases 2013–2020 in Cambodia, evidence-based national diabetes guidelines are not available in the country. Primary care facilities rarely maintain stocks of diagnostic tests related to screening or diabetic drug therapies and are generally unable to provide treatments for diabetes-related complications [3]. Strategic investments in the strengthening of health systems for diabetes could mean far-reaching positive impacts on the quality and effectiveness of primary health care services as well as indirectly strengthen capacity to tackle communicable diseases. Programs for diabetes care and management have focused on similar general features, including increasing access to screening and laboratory testing, appropriate outpatient services (i.e., individualized patient consultations and/or counseling), and prescribed routine medication [13,14]. The Health Equity and Quality Improvement Project (H-EQIP), which aims to improve access to quality health services in Cambodia, has dedicated funds to enable services for diabetes and hypertension [15]. These funds, disbursed and monitored by the World Bank, require that district-level health facilities have appropriate rooms, trained staff, equipment, adequate supplies and drug availability, referral and monitoring system, and community structure to support home-based care for uncomplicated cases in accordance with guidelines. Funds also reward increases in screening coverage.

### Role for public social protection schemes in diabetes care and management

Providing sufficient financial risk protection from routine health care expenditures may be part of the solution towards effectively tackling the diabetes burden and motivating individuals to appropriately seek care to effectively manage their condition. In Cambodia, the Health Equity Funds (HEF) act as the main social health protection scheme [16,17]. Established in 2000 to reduce barriers to accessing care, HEF funds user fee exemptions at public health facilities for the poor [16]. Diabetes screening and drug therapies are included as essential services and medicines in the Guidelines on Minimum Package of Services for Health Center Development distributed by the Ministry of Health (MOH) [18,19]. Yet, HEF offers only limited financial access to medicines for patients with diabetes, particularly in rural areas [20]. Instead,

diabetes patients rely heavily on private health services, known to drive high out-of-pocket (OOP) expenditures and contribute to negative consequences on access and adherence to treatment [20].

Benefit design of health insurance and other social protection schemes has been shown to influence patient and provider behaviors, affecting health services utilization and provision [21–25]. Studies on the impact of user fee exemptions for health services on health care utilization have shown mixed results depending on target population and types of services [26–29]. In the case of diabetes care, in theory, if cost remains a barrier to accessing particular services, such as fasting plasma glucose (FPG) tests and medications, reducing these barriers may encourage their utilization. If providers expect to receive sufficient reimbursement for the provision of diabetes-related services, they may be incentivized to offer such services [30–32]. In this way, benefit design of essential health services packages and social protection schemes can have an impact on diabetes diagnosis, appropriate treatment, and glucose control for the prevention of disease complications.

## Study objective

We estimate the health and financial impacts of strategies providing financial coverage for diabetes care through HEF in Cambodia. As a first step, we focus on how increased availability of such services would impact the poorest in the population, since these households are most at risk for catastrophic health expenditures. We also highlight the differential impacts of strategies on men and women that may be important for appropriate policy design.

## Methods

### Study setting

Cambodia is a lower middle-income country (GDP per capita: 2017 USD 1,384) in Southeast Asia with a population of approximately 16 million, 80% of which live in rural areas [16,33]. Life expectancy was 67 and 71 years for men and women, respectively, and under-five mortality was 28 per 1,000 live births in 2018 [34,35]. Despite considerable economic growth in the past two decades, Cambodia's poverty rate remains high: about 18% of the Cambodian population was living under the international poverty line ($1.90 per day, 2011 purchasing power parity) in 2014 [36]. Total health expenditure was estimated to be about US$78 per capita in 2016, financed by individuals' OOP payments (60%), the government (20%), and donors (20%) [37,38]. Total health expenditure was 6% of GDP in 2016, below the average 9% of the Southeast Asia region, with government spending accounting for 22% of that expenditure (corresponding to 1.3% of GDP) [33,39].

### Integrating diabetes trajectory and care delivery in a Markov model

We designed a Markov microsimulation to compute costs and health and economic outcomes related to diabetes care and management for a cohort aged 25 to 69 years in Cambodia using a societal perspective (Fig 1) [40,41]. The model of care follows a general structure of patients undergoing diabetes screening, receiving drug therapies, and receiving treatment for related complications. The model simulated 45 years of individuals' lives. For each annual cycle of the model, a Markov chain was constructed based on an individual's characteristics and transition probabilities between disease states. New individuals were not introduced over the course of the model's time horizon.

Initial states characterizing the population were based on current population distributions for sex, age, income, and diabetes burden reported by the 2014 Demographic and Health

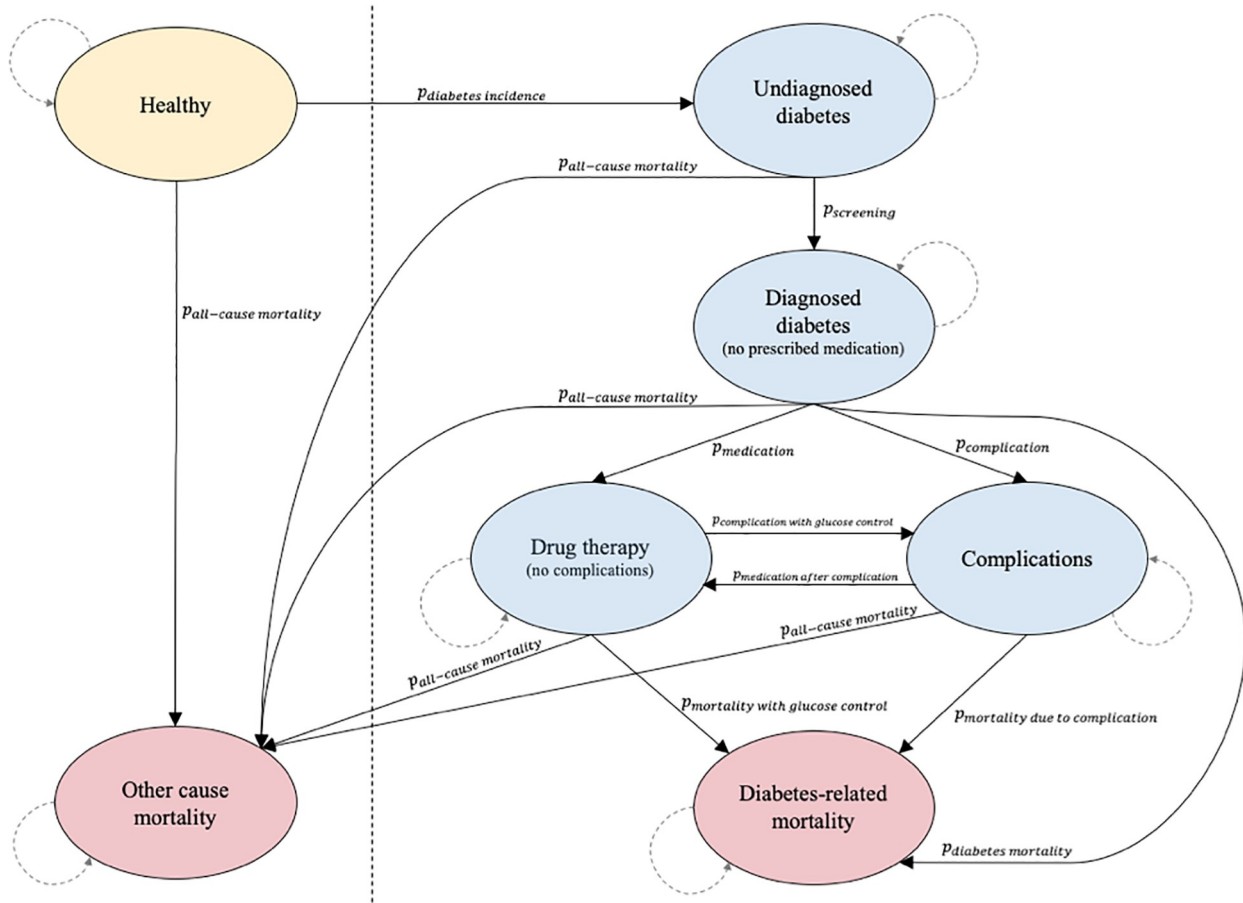

**Fig 1. Representation of the Markov model integrating disease and care delivery states.** 'Drug therapy' state includes oral anti-diabetic drug therapy and insulin therapy options. 'Complications' state entails incidence of major complications and subsequent treatment.

Survey, 2017 Cambodian Socio-Economic Survey, and the 2017 Global Burden of Disease (GBD) Study [5,42,43]. The WHO Global Health Observatory reports age- and sex-specific all-cause mortality for Cambodia in five-year age groups; mortality was assumed to be constant across each five-year interval by which data were available and did not evolve for age or epidemiological changes over the course of the model's time horizon [39]. A gamma distribution was used to simulate individuals' annual income in the country, informed by the national average monthly income for 2017 alongside the Gini coefficient of 36.0 for 2013 [43,44]. Disability weights for diabetes-related states were obtained from the GBD Study 2017 [5].

The model specifies care seeking behavior based on data available for household care seeking for the most recent illness as well as the type of facility visited [45,46]. For an individual to be officially diagnosed with diabetes, an individual is required to have visited a health facility, been screened with an FPG test, and received laboratory testing results. Probability of treatment with oral anti-diabetic drug (OAD) or insulin therapies was informed by the proportion of diabetes patients on medication prior to enrollment in a diabetes management program in Cambodia [14]. In the model, it is possible to be diagnosed with diabetes and remain untreated; this represents cases in which diabetes has been diagnosed and the provider gives a diet consultation rather than drug prescription. The existing literature remains inconclusive regarding how likely diabetic patients successfully improve their diets or other lifestyle habits.

Thus, the model conservatively assumes only a small reduction in relative risk of subsequent complications (RR = 0.90). The effects of glucose control on mortality and/or incidence of complications were understood as representative of the impact of either OAD or insulin regimens. Because medication adherence is a significant factor in successful control of glucose levels, degree of OAD or insulin adherence is also considered in the model. Parameters representing OAD were derived from information on metformin, the likely candidate for OAD therapies in this setting.

An individual may die due to complications induced by the symptoms or effects of diabetes. Eight complications related to diabetes were considered: (i) nephropathy, (ii) retinopathy, (iii) neuropathy, (iv) angina pectoris, (v) peripheral vascular disease (PVD), (vi) myocardial infarction (MI), (vii) stroke, and (viii) heart failure. These complications were selected as the main diabetes-related complications based on the published literature and past costing studies for diabetes focused on Cambodia [7,47]. We conservatively assumed that complication-related states represent the most severe stages of a complication. Complication-specific costs were included based on the most common and/or feasible treatment in Cambodia.

HEF eligibility was assumed for individuals with an annual income of below approximately US$377, which corresponds to the 20th percentile of Cambodia's estimated national income distribution [48]. Because the specific requirements for HEF eligibility are not released publicly, the exact eligibility threshold is not known. For this reason, the model was also run for an eligibility threshold at the 30th percentile of the estimated income distribution. Upon seeking care at public facilities, about 88% of HEF beneficiaries claim their benefits under the scheme [49]. The model assumed that changes in care seeking behavior would be driven by the knowledge of HEF coverage and of availability of diabetes-related services in public facilities. Given the high costs generally associated with diabetes management, we considered two different scenarios of HEF coverage of direct medical costs associated with diabetes-related care to illustrate the range of potential impact of HEF strategies. In the first case, HEF is assumed to provide financial coverage for 80% of direct medical costs that would otherwise be paid out-of-pocket. In the second case, HEF coverage is assumed to cover 100% of those direct medical costs.

## Health coverage strategies for diabetes care

Six strategies involving changes to HEF coverage for diabetes care were compared to the current standard in which HEF beneficiaries do not receive any financial coverage for available diabetes-related services (Table 1). These strategies were selected based on the cascade of care for diabetes and how service packages are defined in HEF (see S1 Appendix). Strategies varied in terms of which segments of the diabetes cascade of care would be covered: diagnostics, medications, and treatment of diabetes-related complications. Evidence suggests that HEF reduces the probability of seeking care in the private sector and shows that public health services utilization increases among HEF beneficiaries compared to "near-poor" non-beneficiaries (i.e., individuals just above the threshold of HEF eligibility) [20,45,49–54]. For scenarios in which services for each strategy would be available in the public sector, we assumed that HEF coverage for these services would increase their utilization among HEF beneficiaries.

Strategies that included the coverage of diagnostic services were assumed to impact the probabilities of care seeking behavior for diabetes-related symptoms and subsequent diagnosis in the presence of disease. Strategies involving medications were assumed to impact the probabilities of drug adherence for diabetes and strategies related to complications of the disease were assumed to affect the probability of care seeking for complications. While coverage for diabetes-related complications may compel patients to seek care, there may be little capacity of

**Table 1. Description of strategies analyzed and their hypothetical impacts.**

| Strategy | HEF Coverage | Impact |
| --- | --- | --- |
| Current standard | No effective financial coverage for any diabetes-related services | Represents current situation |
| Diagnostics only | Diagnostic services for diabetes, including screening and laboratory testing | Reduces barriers to care seeking, increasing probability of diagnosis (RR = 1.5) and care utilization (RR = 2.0) |
| Drug therapy only | Prescribed medication (i.e., oral anti-diabetic (OAD) medication and/or insulin) | Reduces barriers to access to medicines, increasing drug adherence ($p = 0.40$) |
| Complications only | Treatment for diabetes-related complications | Reduces barriers to care seeking, increasing probability of care utilization (RR = 2.0) |
| Diagnostics + Drug therapy | Diagnostic services and prescribed medication | Combined effect of strategies for diagnostics and drug therapy |
| Drug therapy + Complications | Prescribed medication and treatment for diabetes-related complications | Combined effect of strategies for drug therapy and complications |
| Diagnostics + Drug therapy + Complications | Diagnostic services, prescribed medication and treatment for diabetes-related complications | Combined effects of strategies for diagnostics, drug therapy, and complications |

RR: Relative risk; *p* indicates a probability value.

Impacts described above were simulated in the model as relative risks or changes in probability to represent assumed behavior changes among HEF beneficiaries.

facilities to provide curative care for severe acute conditions, such as myocardial infarction or stroke, and services may minimally reduce mortality. Additionally, there is little data available to inform how formal treatment of complications affect health outcomes in Cambodia. As a result, the potential effects of health care seeking and treatment for complication-related mortality were not accounted for in the model.

## Computing health gains, public costs, and financial risk protection

A literature review supplied the information for each input parameter included in the model (Table 2). Whenever possible, parameters were specific to Cambodia or similar settings.

Disability-adjusted life-years (DALYs) were used to capture health impacts. DALYs were calculated as the sum of years of life lost (YLL) due to premature mortality and years of life lived with disability (YLD) among people living with diabetes and its consequences [55,56]. YLLs were calculated using the life expectancy for Cambodia at the age at which death occurred. YLDs were estimated for each yearly iteration of the model as the number of individuals in each state multiplied by the corresponding disability weight for that state. The disability weight of 0.049 associated with uncomplicated diabetes was incurred upon being diagnosed with diabetes in the model. DALYs were not discounted over time [5,57].

Costs of screening, laboratory testing, OAD and insulin therapies, facility-based treatments for each complication, outpatient visits, and hospitalization with an average length of stay of 5 days were included in the model [46]. Non-medical costs included transportation to outpatient visits or for hospitalizations [20,45]. All costs data were converted into 2019 USD and discounted using a 3% annual discount rate [58].

The extent of financial risk protection across the various HEF coverage scenarios was measured by individuals' direct medical and non-medical costs, the incidence of catastrophic health expenditures (CHE), and the number of households pushed below the international poverty line as a result of OOP expenditures related to diabetes (i.e., impoverishment) [59]. A case of CHE would be incurred when OOP expenditures exceeded 40% of household income; thresholds of 10% and 25% were also examined [60,61]. Because much of the poorest 20% of the Cambodian population is already below the international poverty line, impoverishment was relevant when the poorest 30% of the population was eligible for HEF coverage. Results

**Table 2. Input parameters used in the economic evaluation of coverage of diabetes-related services under Health Equity Funds in Cambodia.**

| Parameter | Description | Value | Probability Distribution | Source |
|---|---|---|---|---|
| **Costs*** | | | | |
| Diagnostics | Unit cost of fasting plasma glucose (FPG) test | 1.10 | *Lognormal*(−0.232, 0.809) | Flessa & Zembok, 2014 |
| Laboratory | Cost of laboratory services for diagnostic testing | 1.41 | *Lognormal*(−0.024, 0.857) | Flessa & Zembok, 2014 |
| Oral anti-diabetic (OAD) therapy | Annual average cost of OAD per patient | 27.86 | *Lognormal*(3.245, 0.406) | Flessa & Zembok, 2014 |
| Insulin | Annual average cost of insulin per patient | 125.80 | *Lognormal*(4.816, 0.195) | Flessa & Zembok, 2014 |
| Outpatient visits | | | | |
| *Health center* | Cost of outpatient visit to health center | 4.15 | *Lognormal*(1.299, 0.496) | Flessa *et al.*, 2018 |
| *CPA1 facility* | Cost of outpatient visit to CPA1 facility | 10.32 | *Lognormal*(2.298, 0.268) | Flessa *et al.*, 2018 |
| *CPA2 facility* | Cost of outpatient visit to CPA2 facility | 6.28 | *Lognormal*(1.754, 0.408) | Flessa *et al.*, 2018 |
| *CPA3 facility* | Cost of outpatient visit to CPA3 facility | 44.40 | *Lognormal*(3.791, 0.059) | Flessa *et al.*, 2018 |
| Inpatient visits (per day) | | | | |
| *Health center* | Cost of inpatient visit to health center | 4.77 | *Lognormal*(1.479, 0.408) | Flessa *et al.*, 2018 |
| *CPA1 facility* | Cost of inpatient visit to CPA1 facility | 59.73 | *Lognormal*(4.083, 0.115) | Flessa *et al.*, 2018 |
| *CPA2 facility* | Cost of inpatient visit to CPA2 facility | 29.53 | *Lognormal*(3.378, 0.115) | Flessa *et al.*, 2018 |
| *CPA3 facility* | Cost of inpatient visit to CPA3 facility | 40.85 | *Lognormal*(3.703, 0.115) | Flessa *et al.*, 2018 |
| Discount rate | Discount rate applied to public spending | 0.03 | - | Assumption |
| **Individual expenditures*** | | | | |
| Transport costs | | | | |
| *Outpatient care* | Average cost of transport for care seeking at a health center | 0.92 | *Lognormal*(−0.138, 0.330) | Jacobs *et al.*, 2018 |
| *Inpatient care* | Average cost of transport for care seeking at a public hospital | 11.65 | *Lognormal*(2.453, 0.068) | Jacobs *et al.*, 2018 |
| **Population characteristics** | | | | |
| All-cause mortality | Age- and sex-adjusted all-cause mortality | 0.046 | - | WHO, 2019 |
| Diabetes incidence (mean) | Age- and sex-adjusted annual diabetes incidence | 0.001 | - | IHME, 2017 |
| Diabetes prevalence (mean) | Age- and sex-adjusted diabetes prevalence | 0.062 | - | IHME, 2017 |
| Diabetes-related mortality | Age- and sex-adjusted diabetes-related mortality | 0.030 | - | IHME, 2017 |
| Income | Average annual household income (USD, 2017) | 5,783 | *Gamma*(0.5, 11566.32) | MOP/NIS, 2019 |
| **Diabetes-related complications** | | | | |
| Undiagnosed diabetes | | | | |
| *Disability weight* | Disability weight for undiagnosed diabetes | 0.049 | - | IHME, 2017 |
| Nephropathy | | | | |
| *Incidence* | Annual incidence of neuropathy among diabetics | 0.0100 | - | Gheith *et al.*, 2016 |
| *Effectiveness of therapy* | Effect of glucose-lowering agents on incidence of nephropathy (RR) | 0.30 | - | Chaudhury *et al.*, 2017; UKPDS 34, 1998; UKPDS 33, 1998 |
| *Disability weight* | Disability weight for nephropathy (stage 5) | 0.569 | - | IHME, 2017 |

(*Continued*)

**Table 2.** (Continued)

| Parameter | Description | Value | Probability Distribution | Source |
|---|---|---|---|---|
| *Case fatality rate* | Nephropathy-related mortality among diabetics | 0.311 | - | Afkarian *et al.*, 2013 |
| *Treatment cost* | Annual cost of nephropathy treatment (outpatient, hemodialysis) | 6,358 | *Lognormal*(8.733, 0.219) | Mushi *et al.*, 2015 |
| Retinopathy | | | | |
| *Incidence* | Annual incidence of retinopathy among diabetics | 0.0212 | - | Ahmed *et al.*, 2012 |
| *Effectiveness of therapy* | Effect of glucose-lowering agents on incidence of retinopathy (RR) | 0.68 | - | UKPDS 33, 1998 |
| *Disability weight* | Disability weight for retinopathy | 0.184 | - | IHME, 2017 |
| *Case fatality rate* | Retinopathy-related mortality among diabetics | 0 | - | Assumption |
| *Treatment cost* | Annual cost (USD) of retinopathy treatment (outpatient, intravitreal injection) | 330 | *Lognormal*(5.792, 0.115) | Sasongko *et al.*, 2019 |
| Neuropathy | | | | |
| *Incidence* | Annual incidence of neuropathy among diabetics | 0.0466 | - | Sands *et al.*, 1997 |
| *Effectiveness of therapy* | Effect of glucose-lowering agents on incidence of neuropathy (RR) | 0.94 | - | Juster-Switlyk & Smith, 2016 |
| *Disability weight* | Disability weight for neuropathy | 0.133 | - | IHME, 2017 |
| *Case fatality rate* | Neuropathy-related mortality among diabetics | 0 | - | Assumption |
| *Treatment cost* | Daily cost (USD) of acetylsalicylic acid (outpatient) | 2.62 | *Lognormal*(6.862, 0.029) | WHO/HAI, 2015 |
| Angina pectoris | | | | |
| *Incidence* | Annual incidence of angina pectoris among diabetics | 0.0067 | - | UKPDS 33, 1998 |
| *Effectiveness of therapy* | Effect of glucose-lowering agents on incidence of angina pectoris (RR) | 0.68 | - | UKPDS 33, 1998 |
| *Disability weight* | Disability weight for angina pectoris (moderate) | 0.080 | - | IHME, 2017 |
| *Case fatality rate* | Angina pectoris-related mortality among diabetics | 0 | - | Assumption |
| *Treatment cost* | Daily cost (USD) of beta blocker (outpatient) | 7.07 | *Lognormal*(7.855, 0.29) | WHO/HAI, 2015 |
| Peripheral vascular disease (PVD) | | | | |
| *Incidence* | Annual incidence of PVD among diabetics | 0.0085 | - | Mata-Cases *et al.*, 2011 |
| *Effectiveness of therapy* | Effect of glucose-lowering agents on incidence of PVD (RR) | 0.74 | - | UKPDS 33, 1998 |
| *Disability weight* | Disability weight for PVD | 0.014 | - | IHME, 2017 |
| *Case fatality rate* | Probability of amputation due to PVD and subsequent death | 0.002 | - | Hoffstad *et al.*, 2015 |
| *Treatment cost* | Daily cost (USD) of beta blocker (outpatient) | 7.07 | *Lognormal*(7.855, 0.29) | WHO/HAI, 2015 |
| Myocardial infarction (MI) | | | | |
| *Incidence* | Annual incidence of MI among diabetics | 0.174 | - | UKPDS 33, 1998 |
| *Effectiveness of therapy* | Effect of glucose-lowering agents on incidence of MI (RR) | 0.39 | - | Chaudhury *et al.*, 2017; UKPDS 34, 1998 |
| *Disability weight* | Disability weight for MI | 0.432 | - | IHME, 2017 |
| *Case fatality rate* | MI-related mortality among diabetics | 0.707 | - | UKPDS 34, 1998 |
| *Treatment cost* | Daily cost (USD) of beta blocker (inpatient) | 7.07 | *Lognormal*(7.855, 0.29) | WHO/HAI, 2015 |
| Stroke | | | | |
| *Incidence* | Annual incidence of stroke among diabetics | 0.0053 | - | UKPDS 33, 1998 |
| *Effectiveness of therapy* | Effect of glucose-lowering agents on incidence of stroke (RR) | 0.59 | - | UKPDS 33, 1998 |

(*Continued*)

**Table 2.** (Continued)

| Parameter | Description | Value | Probability Distribution | Source |
|---|---|---|---|---|
| *Disability weight* | Disability weight for stroke (level 5) | 0.588 | - | IHME, 2017 |
| *Case fatality rate* | Stroke-related mortality among diabetics | 0.693 | - | Baena-Díez *et al.*, 2016 |
| *Treatment cost* | Daily cost (USD) of acetylsalicylic acid (inpatient) | 2.62 | *Lognormal*(6.862, 0.029) | WHO/HAI, 2015 |
| Heart failure | | | | |
| *Incidence* | Annual incidence of heart failure among diabetics | 0.0033 | - | UKPDS 33, 1998 |
| *Effectiveness of therapy* | Effect of glucose-lowering agents on incidence of heart failure (RR) | 0.68 | - | UKPDS 33, 1998 |
| *Disability weight* | Disability weight for heart failure (severe) | 0.179 | - | IHME, 2017 |
| *Case fatality rate* | Heart failure-related mortality among diabetics | | - | Baena-Díez *et al.*, 2016 |
| *Treatment cost* | Daily cost (USD) of beta blocker (inpatient) | 7.07 | *Lognormal*(7.855, 0.29) | WHO/HAI, 2015 |
| **Care delivery & utilization** | | | | |
| Diagnosis | Percentage of diabetics with previous diagnosis by provider | 0.370 | *Beta*(8.241, 14.057) | Oum *et al.*, 2010 |
| Care seeking | | | | |
| *Outpatient visit (non-HEF)* | Probability of outpatient care utilization (public) for diabetes complications | 0.117 | *Beta*(0.916, 4.412) | Nagpal *et al.*, 2019 |
| *Outpatient visit (HEF)* | Probability of outpatient care utilization (public) for diabetes complications | 0.172 | *Beta*(0.420, 3.171) | Nagpal *et al.*, 2019 |
| *Inpatient visit* | Probability of hospitalization (public) for diabetes complications | 0.015 | *Beta*(8.850, 581.150) | Nagpal *et al.*, 2019 |
| Provider choice | | | | |
| *Health center* | Proportion seeking treatment from health center for most recent illness | 0.114 | *Beta*(9.755, 107.796) | NIS/ICF, 2018 |
| *CPA1 facility* | Proportion seeking treatment from CPA1 facility for most recent illness | 0.025 | *Beta*(10.943, 574.050) | NIS/ICF, 2018 |
| *CPA2 facility* | Proportion seeking treatment from CPA2 facility for most recent illness | 0.030 | *Beta*(11.862, 557.843) | NIS/ICF, 2018 |
| *CPA3 facility* | Proportion seeking treatment from CPA3 facility for most recent illness | 0.042 | *Beta*(7.304, 190.218) | NIS/ICF, 2018 |
| Drug therapies | | | | |
| *OAD* | Probability of receiving OAD prescription | 0.224 | - | Taniguchi *et al.*, 2017 |
| *Insulin* | Probability of receiving insulin prescription | 0.017 | - | van Olmen *et al.*, 2016 |
| *Combination* | Probability of receiving OAD + insulin prescription | 0.07 | - | van Olmen *et al.*, 2016 |
| *OAD to insulin* | Probability of transitioning from OAD to insulin therapy | 0.04 | - | Ringborg *et al.*, 2010 |
| *Adherence* | Probability of adhering to prescribed therapy | 0.125 | - | Assumption, Flessa & Zembok, 2014 |
| **Health Equity Funds (HEF)** | | | | |
| Eligible | Income percentile eligible for HEF | 0.20, 0.30 | - | Assumption |
| Enrolled | Proportion of target population enrolled in HEF | 0.75 | - | Annear *et al.*, 2015 |
| Coverage benefits | Proportion of expenditures covered under HEF (subsidy rate) | 0.80, 1.00 | - | Assumption |
| Utilization | Proportion of beneficiaries using HEF at point-of-service | 1.00 | - | Assumption |

*All monetary values were adjusted to 2019 USD.

All cost-related parameters describe the cost per instance (i.e., per visit, per use of transport, etc.) unless otherwise stated (e.g., annual costs of drug therapies).

All disease-related parameters refer to annual estimates.

are reported as incremental cost-effectiveness ratios (ICERs) in terms of cost per DALY averted and cost per CHE case averted. Outcomes of interest were computed based only on utilization of services among diabetics.

A population of 800,000 was simulated to represent the estimated 16 million people of Cambodia's population. Analyses were limited to only those individuals who would be considered eligible for HEF and thereby potentially receive the intervention. Thus, the model was implemented on the subsample of 160,000 individuals who were the poorest 20% and who were assumed to be eligible for HEF (240,000 individuals for the poorest 30% eligible) per comparator strategy. The resulting costs, OOP expenditures, and financial risk protection estimates were then inflated to represent this subpopulation, about 3.2 million people likely eligible for HEF in Cambodia.

To assess the robustness of our findings, uncertainty analyses of parameters and the sample population were conducted. Parameters for costs and effectiveness of strategies were varied to generate 50 distinct sets of parameters drawn from defined probability distributions. Lognormal distributions were applied to represent potential variability in costs and beta distributions to represent potential variability in probabilities used in the model. Effects of HEF strategies were expressed as relative risks in the model and varied according to a normal distribution (Table 1, $\sigma^2 = 0.1$).

To test the uncertainty of results based on sample characteristics, 40,000 individuals were sampled with replacement from the simulated population to generate 100 distinct populations. Outcomes of interest were computed for each of these 100 populations to gauge the range of possible results given the potential variability in parameters and population characteristics in Cambodia.

All analyses were conducted in R (version 3.6.1).

### Ethical approval

Patients or the public were not involved in the design, conduct, reporting, or dissemination plans of this research.

## Results

### State duration

Strategies affected the duration of time individuals spent in model states. Those strategies providing coverage for diagnostic services resulted in a greater number of diagnosed cases and, subsequently, more individuals receiving drug therapy options. Strategies with coverage of medications resulted in less time spent in states of diabetes-related complications. This time instead was spent in states of diagnosed diabetes, either being prescribed diet and/or lifestyle advice or drug therapy. Providing coverage for both diagnostic services and medications amplified these impacts, decreasing the level of undiagnosed diabetes and increasing the number individuals receiving drug therapies.

Due to lack of data, we did not account for the potential impact of treatment of severe complications to prevent subsequent mortality. Thus, HEF coverage for complications did not result in any changes to the duration of time spent in disease states. Across strategies, women had more diabetes-related cardiovascular conditions than their male counterparts. In combination with higher diabetes mortality for women, this drove greater health impacts among women compared to men.

### Costs associated with HEF coverage of diabetes services

HEF coverage for the full continuum of care for diabetes resulted in the highest costs, while providing coverage for solely drug therapy resulted in the lowest costs per HEF beneficiary.

**Table 3. Incremental costs and health impact by strategy and HEF coverage scenario.**

| Strategy | HEF Eligibility | Total Costs | Incremental Costs | DALYs | Incremental DALYs | ICER (US$/DALY averted) |
|---|---|---|---|---|---|---|
| Current standard | 20% | 222,241,881 | - | 2,685,856 | - | - |
| | 30% | 309,847,914 | - | 4,001,523 | - | - |
| Diagnostics only | 20% | 375,570,938 | 153,329,057 | 2,695,079 | 9,222 | - |
| | 30% | 462,913,640 | 153,065,726 | 4,010,745 | 9,222 | - |
| Drug therapy only | 20% | 223,283,447 | 1,041,566 | 2,647,453 | -38,404 | 27 |
| | 30% | 308,635,759 | -1,212,155 | 3,963,119 | -38,404 | -32 |
| Complications only | 20% | 376,546,924 | 154,305,043 | 2,685,856 | 0 | - |
| | 30% | 465,388,656 | 155,540,742 | 4,001,523 | 0 | - |
| Diagnostics + Drug therapy | 20% | 365,140,541 | 142,898,661 | 2,647,515 | -38,341 | 3,727 |
| | 30% | 451,558,506 | 141,710,592 | 3,963,181 | -38,341 | 3,696 |
| Drug therapy + Complications | 20% | 365,438,949 | 143,197,068 | 2,647,453 | -38,403 | 3,729 |
| | 30% | 451,167,284 | 141,319,370 | 3,963,119 | -38,404 | 3,680 |
| Diagnostics + Drug therapy + Complications | 20% | 658,879,070 | 436,637,189 | 2,647,515 | -38,341 | 11,388 |
| | 30% | 747,217,986 | 437,370,072 | 3,963,181 | -38,341 | 11,407 |

HEF: Health Equity Funds; DALY: Disability-adjusted life-year; ICER: Incremental cost-effectiveness ratio.

Note: Incremental values have been computed for each strategy compared to the current standard.

Examination of incremental costs between strategies and the current provision of care showed that coverage of drug therapies was cost-saving due to its prevention of high-cost severe complications (Table 3). Treatment for diabetes-related complications consistently comprised the greatest proportion of costs. Since covering medication costs resulted in reduced complications, costs were reduced for this strategy, mitigating any initial cost increases due to increased coverage of diagnostic services. Nevertheless, the costs of treating diabetes-related complications as a result of increased care seeking for diabetes care drove increasing incremental costs.

The mean annual direct medical cost without intervention was estimated to be about $62 per diabetic patient. Adopting any strategy would significantly reduce the direct medical costs incurred by diabetic patients over 45 years. HEF coverage, to be paid for by the government, would amount to between $28 and $58 per diabetic patient depending on the extent of coverage and services covered. The magnitude of total direct medical costs over time changed as a function of the number of individuals covered by HEF.

## Health impact

Strategies showed varied impacts on health (Fig 2). Coverage for only diagnostic services resulted in increased DALYs in the population because diagnosed, uncomplicated diabetes incurs disability (0.049) due to worry and interference with daily activity [5]. Covering only diagnostic services was the only strategy that would result in high expenditures for DALYs incurred. Without corresponding increases in appropriate care and management of the disease, there would be no means by which to prevent the development of later complications that result in significant DALYs. On the other hand, treatment of diabetes through regular drug therapy would result in DALYs averted that can compensate for the DALYs incurred upon diagnosis of uncomplicated diabetes. Covering drug therapies under HEF would cost $27 per DALY averted if the poorest 20% of the population were eligible for HEF, while covering diagnostic services as well would cost an estimated $2,468 per DALY averted. Providing coverage for drug therapies and treatment for complications would cost about $3,727 per DALY averted.

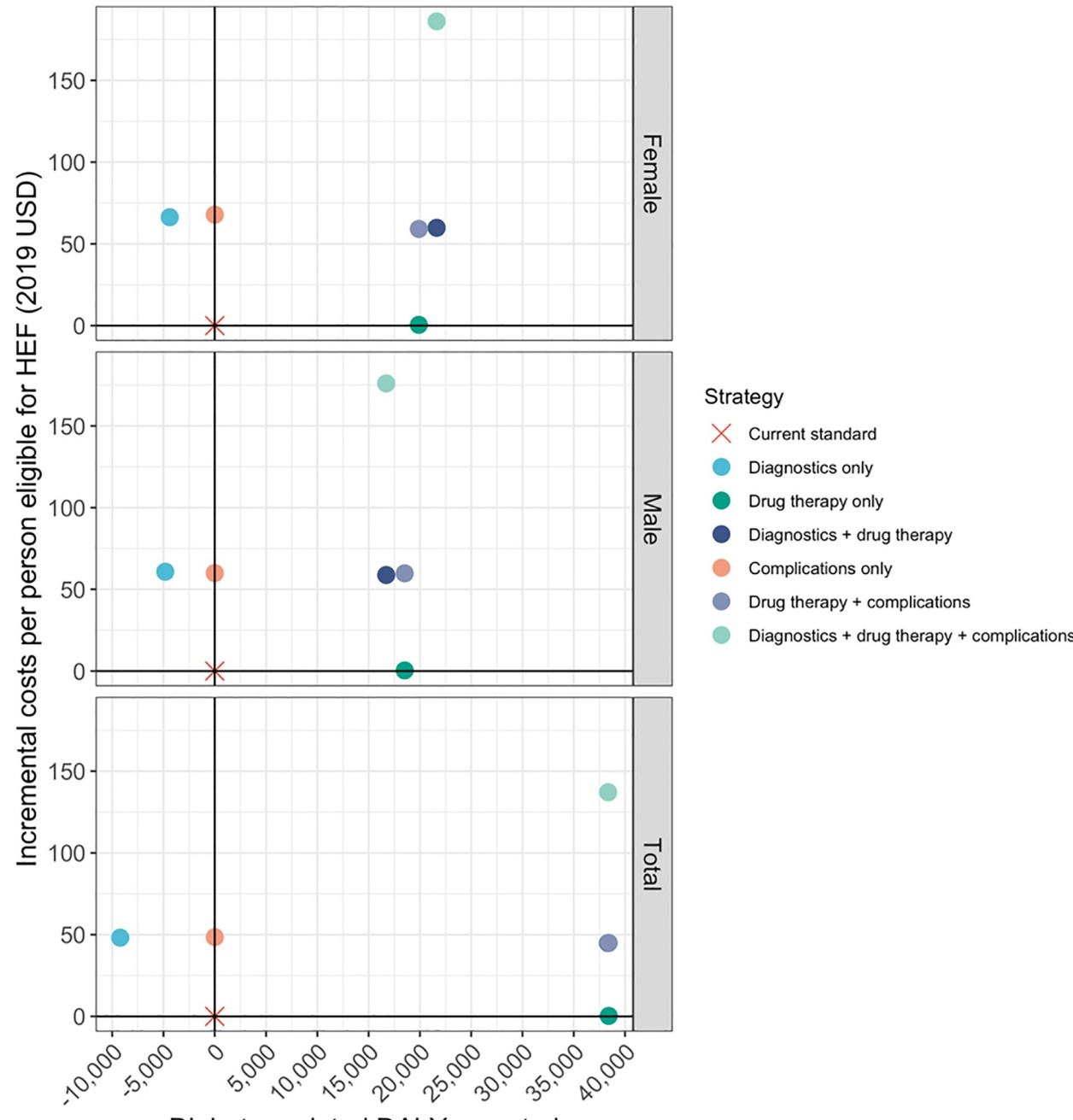

**Fig 2. Incremental total costs per person eligible for HEF (poorest 20%) vs. diabetes-related DALYs averted by population group and strategy.**

Examination of the states that drove DALYs averted upon coverage of diabetes medications revealed that about 36% of the DALYs averted would be due to prevention of diabetes-related death. Coverage for both diagnostic services and drug therapies contributed to the greatest DALYs averted because more diabetes-related deaths were prevented as a result. Those strategies that had positive health impacts consistently drove more favorable ICER estimates among women compared to men.

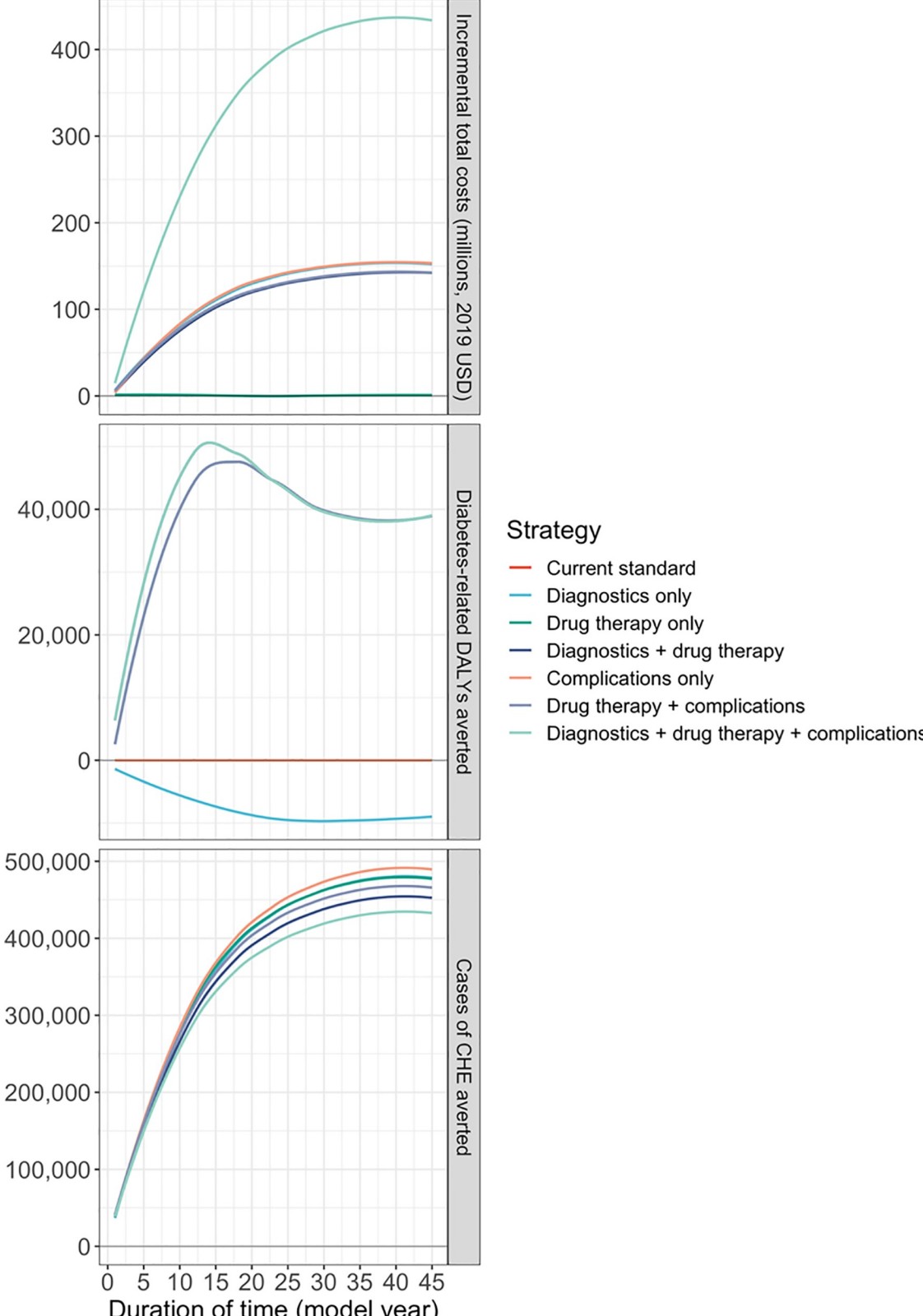

**Fig 3. Incremental costs and impacts over time, HEF eligibility 20%, OOP coverage 100%.**

Cost savings associated with providing coverage for drug therapies could begin as early as the second year of coverage under this scenario. For strategies that provide coverage for complications, model results showed that there would initially be decreases in costs per DALY averted, followed by increases in incremental costs in later years as individuals age, accruing disability from complications and mortality (Fig 3).

### Financial risk protection

According to our model, coverage of drug therapies alone would cost $2 per case of CHE averted (Table 4). Coverage for both diagnostics and drug therapies resulted in the most OOP expenditures averted and would cost an estimated $315 per case of CHE averted. However, coverage of drug therapies and treatment of complications was estimated to cost $306 per case of CHE averted. Providing coverage for all diabetes-related services was the most expensive strategy. Still, about $790 in OOP spending could be averted per diabetic eligible for HEF over

**Table 4. Incremental costs and financial risk protection impact by strategy and HEF coverage scenario.**

| Strategy | HEF Eligibility | OOP Coverage | Total Costs | Incremental Costs | CHE | Incremental CHE | ICER (US$/CHE averted) |
|---|---|---|---|---|---|---|---|
| Current standard | 20% | 100% | 222,241,881 | - | 676,340 | - | - |
| | | 80% | | | | - | - |
| | 30% | 100% | 309,847,914 | - | 746,840 | - | - |
| | | 80% | | | | - | - |
| Diagnostics only | 20% | 100% | 375,570,938 | 153,329,057 | 195,860 | -480,480 | 319 |
| | | 80% | | | 482,720 | -193,620 | 792 |
| | 30% | 100% | 462,913,640 | 153,065,726 | 214,020 | -532,820 | 287 |
| | | 80% | | | 553,100 | -193,740 | 790 |
| Drug therapy only | 20% | 100% | 223,283,447 | 1,041,566 | 197,080 | -479,260 | 2 |
| | | 80% | | | 514,760 | -161,580 | 6 |
| | 30% | 100% | 308,635,759 | -1,212,155 | 213,980 | -532,860 | -2 |
| | | 80% | | | 583,720 | -163,120 | -7 |
| Complications only | 20% | 100% | 376,546,924 | 154,305,043 | 184,900 | -491,440 | 314 |
| | | 80% | | | 459,460 | -216,880 | 711 |
| | 30% | 100% | 465,388,656 | 155,540,742 | 203,440 | -454,320 | 286 |
| | | 80% | | | 529,880 | -117,140 | 717 |
| Diagnostics + Drug therapy | 20% | 100% | 365,140,541 | 142,898,661 | 222,020 | -506,540 | 315 |
| | | 80% | | | 559,200 | -118,460 | 1,220 |
| | 30% | 100% | 451,558,506 | 141,710,592 | 240,300 | -467,640 | 280 |
| | | 80% | | | 628,380 | -151,780 | 1,196 |
| Drug therapy + Complications | 20% | 100% | 365,438,949 | 143,197,068 | 208,700 | -520,640 | 306 |
| | | 80% | | | 524,560 | -152,320 | 943 |
| | 30% | 100% | 451,167,284 | 141,319,370 | 226,200 | -520,640 | 271 |
| | | 80% | | | 594,520 | -152,320 | 928 |
| Diagnostics + Drug therapy + Complications | 20% | 100% | 658,879,070 | 436,637,189 | 241,880 | -434,460 | 1,005 |
| | | 80% | | | 579,720 | -96,620 | 4,519 |
| | 30% | 100% | 747,217,986 | 437,370,072 | 259,640 | -487,200 | 898 |
| | | 80% | | | 649,720 | -97,120 | 4,503 |

HEF: Health Equity Funds; OOP: Out-of-pocket expenditures; CHE: Catastrophic health expenditures; ICER: Incremental cost-effectiveness ratio.

Note: Incremental values have been computed for each strategy compared to the current standard. CHE reported above was measured at the 40% threshold.

the 45-year time horizon in such a scenario. Women benefitted most from coverage of any diabetes-related services across the cascade of care in comparison to men.

## Uncertainty analysis

For four of the six strategies examined, the 95% uncertainty ranges of estimates of health impacts included zero diabetes-related DALYs averted (Fig 4). This suggests that, despite the modeled mean outcomes reported above, there may be populations and/or scenarios under which the health impacts of these strategies are minimal. All of the strategies would reduce the number of CHE cases due to diabetes-related care. Providing coverage for diabetes-related complications consistently reduced the greatest number of CHE cases followed by coverage for diagnostic services and medications. These findings were consistent when analyzing uncertainty under both scenarios of HEF eligibility (20% and 30% of Cambodia's poorest) in the population.

## Discussion

### Summary of key findings

We modeled the benefits of providing financial coverage for diabetes care under HEF for a scenario in which they are indeed available in the public sector. Covering drug therapies would cost $27 per DALY averted and $2 per case of CHE averted over the 45-year time horizon. Coverage that included treatment for complications drove up costs, but prevented CHE, particularly among women. Including diagnostic services in HEF coverage increased negative health impacts as well as costs. Negative health impacts were attributable to the small, but notable, disability weight associated with the psychological worry of having the disease and being on daily medications. Thus, as more individuals are diagnosed with diabetes, more would incur this negative impact on health. However, a benefit package that provided financial coverage for both diagnostic services and diabetes medications would increase the number of individuals receiving appropriate care, thereby increasing positive health impacts, albeit for a cost of $3,727 per DALY averted and $315 per CHE case averted.

This study adopted a societal perspective and, thus, diverged from previous costing analyses conducted for type 2 diabetes in Cambodia in 2014, which estimated costs and health impacts from a government perspective for the country's entire population [7]. Flessa and Zembok estimated $11 million (2013 USD) in costs to the government if diabetes treatment were to be provided to all diabetics in the country [7]. The higher cost estimates can largely be explained by differences in the epidemiological and cost input parameters used.

Study findings align with other costing and cost-effectiveness analyses of diabetes management conducted in LMICs, though there remains wide variation in results across country studies depending on approaches and perspectives [62]. Multiple studies similarly highlight that the most expensive complications were cardiovascular conditions and events, contributing to high annual inpatient costs [62,63]. Intensive glycemic control was reported to be highly cost-effective for the sub-Saharan Africa and Southeast Asia regions in one study [64]. Researchers have also noted varying impacts within patient subgroups, such as among men versus women, emphasizing that policy should carefully weigh differential impacts across sexes [64].

### Financial risk protection and health systems strengthening for diabetes

Simulation of disease states alongside care delivery pathways provided a means to measure the outcomes driven by complex relationships as patients navigate their disease and the health care system in Cambodia. The findings of this study point to opportunities to incur cost

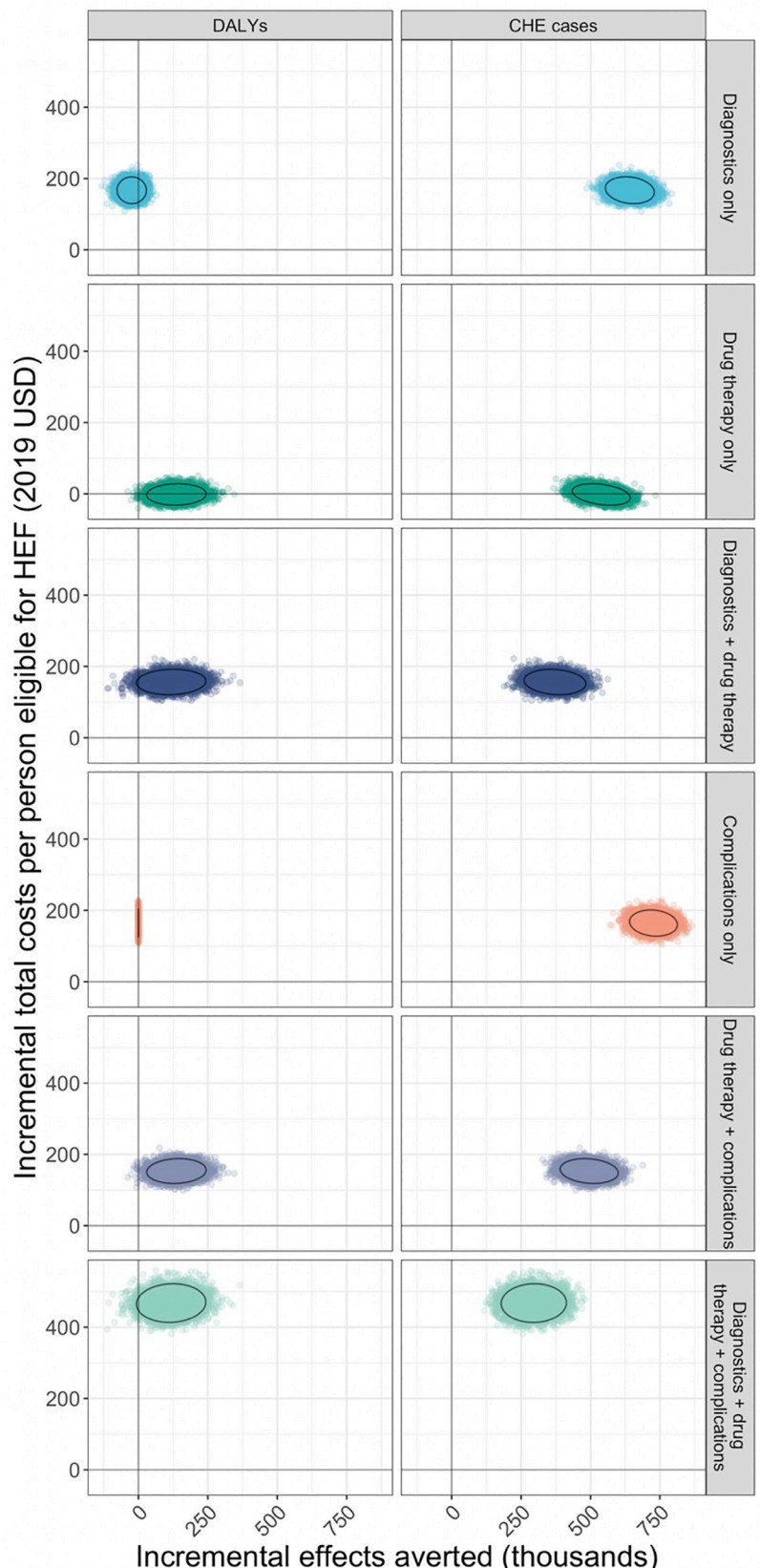

**Fig 4. Results of uncertainty analysis, HEF eligibility 20%. Ellipses indicate 95% uncertainty range of estimates.**

savings and health gains in the effort to control the growing diabetes problem. This study also highlights the burden of diabetes faced by women that extends throughout their lifetimes with the compounding effects of the disease on health and financial security, aligning with the existing literature [15,20,65,66]. Significant impacts for women point to the relevance of targeted interventions for the disease.

Interventions to provide financial coverage for services across the cascade of care for diabetes prompt a parallel focus on improved availability and infrastructure for such services. As is the case in many LMIC settings, Cambodia has not yet established clear practice guidelines for diabetes and it remains unclear how the disease should be managed in the public health care system [67]. Interventions at the level of primary care, the level at which diabetes would ideally be diagnosed and regularly managed, have the potential for far-reaching improvements in regular care and management of diabetes as well as for overall health system strengthening. For example, H-EQIP has been instrumental in propelling efforts to improve the availability of diabetes and hypertension services across levels of the Cambodian public health system, also recognizing the disease's disproportionate impacts on women [15]. Establishing the infrastructure and resources for sufficient diabetes care and services in the public sector is already underway. Exploiting the role of HEF as a mechanism to improve appropriate utilization of screening and diabetes services can be a next cost-effective step to tackling the growing diabetes burden in Cambodia in the long-term.

Because of the complexity of diabetes, a multi-pronged approach that integrates prevention strategies with health system strengthening focused on primary care may be most effective in terms of health gains and costs saved. Indeed, while the focus of this study was on the treatment and management of diabetes, various efforts to prevent diabetes have consistently been shown to be cost-effective. Evidence from this study does not preclude policies for the prevention of diabetes, but rather emphasizes the importance of tackling each step of disease trajectory.

## Limitations and future development of the model

Upon systematic review of type 1 diabetes cost-effectiveness models, Henriksson *et al.* reported that 'best in class' diabetes models employ microsimulation methods to simulate microvascular and macrovascular complications, adverse events, costs, and quality-adjusted life-years (QALYs) over a lifetime time horizon, accounting for parameter uncertainty [68]. The model presented here was a microsimulation that implemented each of these features, though instead computed DALYs [69–72].

Opportunities to improve the model would require enhanced data sources, particularly sources relevant to Cambodia. Data currently available also provides little insight into the impact of outpatient visits and/or hospitalization for diabetes-related complications, such as neuropathy, nephropathy, and cardiovascular conditions. The impacts of formal health care treatment on such complications may be attributable to a complex combination of availability of technology, human resources, health literacy, and quality of care that remains difficult to reliably predict in the Cambodian setting. Additionally, the model does not distinguish between type 1 and type 2 diabetes or early and late diagnosis of diabetes in the population due to lack of data; many of the resources reporting on diabetes in Cambodia describe general diabetes information. The model also does not replenish the population over the course of the model, failing to account for demographic transitions and population aging over time. Yet, aging of the population may have considerable impacts on the epidemiological outcomes for diabetes into the future.

The model in this paper does not consider any startup costs related to the rollout of sufficient service availability for diabetes or any spillover effects on populations beyond those targeted for intervention. However, much of these costs are already being undertaken by

collective development efforts, such as H-EQIP [15]. As the model assumes that the infrastructure and resources are in place for diabetes services at the appropriate level of the health system (i.e., lower levels of care at shorter distances from patient households), we may underestimate current costs of care seeking in which farther travel may be required to reach higher level health facilities. The model does not account for increased utilization of services among those who do not have diabetes. As a result, it is possible that the model underestimates the costs associated with diagnostic services for diabetes. To date, however, there is little information available regarding how often diabetes screening and other diagnostic services are conducted when individuals seek care in Cambodia.

While studies have shown increases in utilization of maternal health services following user fee exemptions, researchers currently have a limited understanding surrounding the magnitude of effects of changes in HEF coverage on patient behaviors, such as care seeking and utilization for diagnostic services and/or treatment for severe complications and drug adherence to OAD and insulin therapies [27]. Further research in these areas would contribute to a more nuanced and precise estimate of costs, health gains, and impacts on financial risk protection. Future modeling to estimate the impact of diabetes coverage policies would benefit from disaggregated data by population subgroups to understand differential impacts across the population and improve equity in access to diabetes-related services [72].

## Implications for policy

Study findings highlight the multifaceted benefits of providing financial coverage for diabetes care, especially for medications for glycemic control. The true benefit of such policies would lie in increased financial risk protection and reduced CHE due to diabetes. Additional benefits could be gained by supporting care seeking for complications, particularly among women, who generally show greater risk of developing CVD.

Equity considerations are important when making decisions on how to design benefit packages for social health protection schemes. Covering drug therapies and treatments for complications could drive further inequities since receiving appropriate diagnostic services could remain a barrier to accessing financial coverage for appropriate treatment. Considering health, financial risk protection, and equity impact together points to outlining a benefit package that, at the very least, provides financial coverage for and availability of diagnostics and medications in a diabetes coverage policy. Adding diagnostic services to the benefit package would cost the government an additional $50 per diabetic HEF enrollee. These costs correspond to $0.20 to $1.41 per capita, which may be reasonable increases in public health spending given that most recent estimates put government health expenditures at only 6% of GDP [33].

Efforts focused on the availability of diabetes-related services under HEF coverage should be informed by detailed information on the costs of medications that the Cambodian public sector would procure for services under the scheme. Similarly, cost estimates would be improved for diagnostic services and treatment for complications if specific unit cost estimates for Cambodia were to become available. This would provide more accurate cost estimations and support informed decision-making with consideration of the burden of diabetes in the long term. Future research should focus on the collection of context-specific data to inform the parameter inputs of simulation models such as the model of this study and, in so doing, inform appropriate strategic planning and management of diabetes for the Cambodian health system.

## Conclusions

Though HEF coverage purports to include screening, testing, and treatment for diabetes, the scheme was ultimately not designed to address the needs of patients with chronic, non-

communicable diseases and assessments have shown that these services remain inaccessible, especially for the poor [20]. Based on the findings of this analysis, efforts to improve service coverage, and subsequently, financial coverage for diabetes diagnostic and treatment therapies for the poor in Cambodia would be highly cost-effective in terms of health and financial risk protection benefits. Women would particularly benefit from availability of and financial risk protection for diabetes-related services under HEF. To improve estimates of costs, health impacts, and financial implications, future models will need more data on care seeking behavior for diabetes as well as the costs of that care seeking, diagnostic and treatment services, and diabetes-related complications. Further research could also explore the impact of improved service and financial coverage for diabetes to the entire Cambodian population.

## Supporting information

**S1 Appendix.**
(DOCX)

## Acknowledgments

The authors would like to thank Thomas J. Bossert, and Sebastian Bauhoff for their valuable support and feedback during the development of this research as well as Michael Law for helpful comments on an earlier version of the paper.

## Author Contributions

**Conceptualization:** Isabelle Feldhaus, Stéphane Verguet.

**Data curation:** Isabelle Feldhaus.

**Formal analysis:** Isabelle Feldhaus.

**Methodology:** Isabelle Feldhaus.

**Project administration:** Isabelle Feldhaus.

**Resources:** Stéphane Verguet.

**Supervision:** Stéphane Verguet.

**Visualization:** Isabelle Feldhaus.

**Writing – original draft:** Isabelle Feldhaus.

**Writing – review & editing:** Isabelle Feldhaus, Somil Nagpal, Stéphane Verguet.

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
