## [Decision Letter · Decision Letter 0]

23 Jul 2021

PONE-D-21-00305

Alleviating the burden of diabetes with Health Equity Funds: Economic evaluation of the health and financial risk protection benefits in Cambodia

PLOS ONE

Dear Dr. Feldhaus,

Thank you for submitting your manuscript to PLOS ONE. After careful consideration, we feel that it has merit but does not fully meet PLOS ONE’s publication criteria as it currently stands. Therefore, we invite you to submit a revised version of the manuscript that addresses the points raised during the review process.

We look forward to receiving your revised manuscript.

Kind regards,

M. Mahmud Khan

Academic Editor

PLOS ONE

Journal Requirements:

We note that one or more of the authors are employed by a commercial company: World Bank Group.

2.1. Please provide an amended Funding Statement declaring this commercial affiliation, as well as a statement regarding the Role of Funders in your study. If the funding organization did not play a role in the study design, data collection and analysis, decision to publish, or preparation of the manuscript and only provided financial support in the form of authors' salaries and/or research materials, please review your statements relating to the author contributions, and ensure you have specifically and accurately indicated the role(s) that these authors had in your study. You can update author roles in the Author Contributions section of the online submission form.

2.2. Please also provide an updated Competing Interests Statement declaring this commercial affiliation along with any other relevant declarations relating to employment, consultancy, patents, products in development, or marketed products, etc.  

Reviewers' comments:

Reviewer's Responses to Questions

**Comments to the Author**

1. Is the manuscript technically sound, and do the data support the conclusions?

Reviewer #1: Partly

2. Has the statistical analysis been performed appropriately and rigorously? 

Reviewer #1: I Don't Know

3. Have the authors made all data underlying the findings in their manuscript fully available?

Reviewer #1: Yes

4. Is the manuscript presented in an intelligible fashion and written in standard English?

Reviewer #1: Yes

5. Review Comments to the Author

Reviewer #1: This is a highly sophisticated study which is not very easy to understand. However:

1. Prevention in high-risk individuals is totally ignored whereas it is critical in order to halt the progression of the diabetes "epidemic" and not overwhelm health services in LMICs. Considering also that there are several studies demonstrating the effectiveness of intensive lifestyle intervention studies in terms of curbing the incidence of diabetes, as well as their cost-effectiveness (see for instance Alouki K et al 2016). Was it not possible to integrate this in the model? If not, at least discuss this major limitation. Even if strengthening the Cambodia health system is on-going, it is doubtful that it could absorb the rising number of diabetes cases and provide quality management.

2. Would the strategies of covering diagnostics only or of covering medication only not also allow for earlier diagnosis and treatment thereby contributing to curbing or delaying complications (Table 1)?

3. That covergae if diagnostics only would increase the negative health impact would have to be better substantiated.

4. The impact of user-fee removal on healthcare seeking has been assessed for conditions other than diabetes. References should be provided on this issue in the discussion, including for instance the study by Beaugé Y et al (2020).

5. Please insert 'diabetes' in the title.

6. PLOS authors have the option to publish the peer review history of their article (what does this mean?). If published, this will include your full peer review and any attached files.

Reviewer #1: No

---

## [Author Response · Author response to Decision Letter 0]

6 Sep 2021

Responses to editor comments: 

1. We have reviewed the PLOS ONE style requirements and revised the manuscript according to the PLOS ONE style template provided in the comments. 

2. One of the authors of the manuscript, SN, is employed by the World Bank Group, whose parent organization is the United Nations. 

2.1. We have provided an amended Funding Statement declaring this affiliation and that the organization did not play a role in the study. Authors' contributions remain the same. SN contributed to the review and final revision of the study. 

2.2. We have provided an updated Competing Interests Statement declaring this affiliation and confirm that it does not alter our adherence to PLOS ONE policies. All authors confirm that we have no competing interests to declare. 

Responses to reviewers' comments:

Reviewer #1: This is a highly sophisticated study which is not very easy to understand. However:

1. Prevention in high-risk individuals is totally ignored whereas it is critical in order to halt the progression of the diabetes "epidemic" and not overwhelm health services in LMICs. Considering also that there are several studies demonstrating the effectiveness of intensive lifestyle intervention studies in terms of curbing the incidence of diabetes, as well as their cost-effectiveness (see for instance Alouki K et al 2016). Was it not possible to integrate this in the model? If not, at least discuss this major limitation. Even if strengthening the Cambodia health system is on-going, it is doubtful that it could absorb the rising number of diabetes cases and provide quality management.

Thank you for this comment and highlighting the importance of prevention measures as some of the most cost-effective efforts to curb the incidence of diabetes. Indeed, we agree. This study focuses on what coverage can be provided by the Health Equity Funds in Cambodia to address the prevalence of diabetes. Unfortunately, prevention measures in the form of lifestyle or other daily activity interventions were determined to be outside of the scope of current HEF coverage policies, which largely focus on services that can be sought at health centers. Discussion of the importance of strategies aimed at preventing diabetes is included in the main text as the last paragraph of the section, “Financial risk protection and health systems strengthening for diabetes”. 

2. Would the strategies of covering diagnostics only or of covering medication only not also allow for earlier diagnosis and treatment thereby contributing to curbing or delaying complications (Table 1)?

Thank you for this insightful comment. Yes, we agree that this is a possibility and a hoped-for impact of such a coverage strategy. Other models on diabetes have stratified cases by early diagnosis and late diagnosis of diabetes. We considered this approach as well. Unfortunately, for Cambodia, there is no information available describing how many diabetes cases are diagnosed early or later in disease progression, and there is limited data on the extent to which early vs. late diagnosis has an impact on treatment outcomes. As a result, we conservatively did not make this distinction in classification of diagnosis. Such an impact may drive more favorable results. Similarly, the typical timing of treatment in the diabetes trajectory remains unknown in Cambodia data sources. We have made revisions in the text to outline these limitations.

3. That coverage if diagnostics only would increase the negative health impact would have to be better substantiated.

The manuscript describes the negative health impact of the diagnostics only strategy due to disability weights attributed to daily psychological stress due to the knowledge of diagnosis result in negative health impact. These disability weights were used because they are standard as reported in the data source (IHME, 2017). In the scenario in which these disability weights are not acceptable for whatever reason, the diagnostics only strategy would have no impact. Following discussion of this point in the discussion section, the primary point made is that treatment subsequent to diagnosis would have truly positive health impacts.

4. The impact of user-fee removal on healthcare seeking has been assessed for conditions other than diabetes. References should be provided on this issue in the discussion, including for instance the study by Beaugé Y et al (2020).

Thank you for your comment and pointing us to this literature. Relevant discussion has been added to the introduction and discussion sections. 

5. Please insert 'diabetes' in the title.

The title of the manuscript already includes the term ‘diabetes’ as follows: “Alleviating the burden of diabetes with Health Equity Funds: Economic evaluation of the health and financial risk protection benefits in Cambodia”.

---

## [Decision Letter · Decision Letter 1]

25 Oct 2021

Alleviating the burden of diabetes with Health Equity Funds: Economic evaluation of the health and financial risk protection benefits in Cambodia

PONE-D-21-00305R1

Dear Dr. Feldhaus,

We’re pleased to inform you that your manuscript has been judged scientifically suitable for publication and will be formally accepted for publication once it meets all outstanding technical requirements.

Kind regards,

M. Mahmud Khan

Academic Editor

PLOS ONE

Additional Editor Comments (optional):

Reviewers' comments:

Reviewer's Responses to Questions

**Comments to the Author**

1. If the authors have adequately addressed your comments raised in a previous round of review and you feel that this manuscript is now acceptable for publication, you may indicate that here to bypass the “Comments to the Author” section, enter your conflict of interest statement in the “Confidential to Editor” section, and submit your "Accept" recommendation.

Reviewer #1: (No Response)

2. Is the manuscript technically sound, and do the data support the conclusions?

Reviewer #1: Yes

3. Has the statistical analysis been performed appropriately and rigorously? 

Reviewer #1: Yes

4. Have the authors made all data underlying the findings in their manuscript fully available?

Reviewer #1: Yes

5. Is the manuscript presented in an intelligible fashion and written in standard English?

Reviewer #1: Yes

6. Review Comments to the Author

Reviewer #1: Our comments were modestly addressed but in the last paragraph before 'Limitations...', references are needed on the cost-effectiveness of prevention through lifestyle interventions. Additionally, the authors will want to consult and eventually cite the studies by Alouki K et al (2015, 2016, 2017) on the medical costs of diabetes treatment in the presence of absence of complications, based on a standardized treatment protocol, in West Africa (and the software for the process).

7. PLOS authors have the option to publish the peer review history of their article (what does this mean?). If published, this will include your full peer review and any attached files.

Reviewer #1: No

---

## [Editor Report · Acceptance letter]

28 Oct 2021

PONE-D-21-00305R1 

Alleviating the burden of diabetes with Health Equity Funds: Economic evaluation of the health and financial risk protection benefits in Cambodia 

Dear Dr. Feldhaus:

I'm pleased to inform you that your manuscript has been deemed suitable for publication in PLOS ONE. Congratulations! Your manuscript is now with our production department. 

Kind regards, 

on behalf of

Dr. M. Mahmud Khan 

Academic Editor

PLOS ONE